

# Family-based whole-exome sequencing identifies novel loss-of-function mutations of *FBN1* for Marfan syndrome

Zhening Pu[1,2,3,*], Haoliang Sun[4,*], Junjie Du[4], Yue Cheng[1,2], Keshuai He[4], Buqing Ni[4], Weidong Gu[4], Juncheng Dai[1,2] and Yongfeng Shao[4]

[1] Department of Epidemiology, School of Public Health, Nanjing Medical University, Nanjing, China
[2] State Key Laboratory of Reproductive Medicine, Nanjing Medical University, Nanjing, China
[3] Center of Clinical Research, Wuxi People's Hospital of Nanjing Medical University, Wuxi, Jiangsu, China
[4] Department of Cardiovascular Surgery, The First Affiliated Hospital of Nanjing Medical University, Nanjing, China
* These authors contributed equally to this work.

Corresponding authors
Juncheng Dai, djc@njmu.edu.cn
Yongfeng Shao, yongfengshao30@hotmail.com

## ABSTRACT

**Background:** Marfan syndrome (MFS) is an inherited connective tissue disorder affecting the ocular, skeletal and cardiovascular systems. Previous studies of MFS have demonstrated the association between genetic defects and clinical manifestations. Our purpose was to investigate the role of novel genetic variants in determining MFS clinical phenotypes.

**Methods:** We sequenced the whole exome of 19 individuals derived from three Han Chinese families. The sequencing data were analyzed by a standard pipeline. Variants were further filtered against the public database and an in-house database. Then, we performed pedigree analysis under different inheritance patterns according to American College of Medical Genetics guidelines. Results were confirmed by Sanger sequencing.

**Results:** Two novel loss-of-function indels (c.5027_5028insTGTCCTCC, p.D1677Vfs*8; c.5856delG, p.S1953Lfs*27) and one nonsense variant (c.8034C>A, p.Y2678*) of *FBN1* were identified in Family 1, Family 2 and Family 3, respectively. All affected members carried pathogenic mutations, whereas other unaffected family members or control individuals did not. These different kinds of loss of function (LOF) variants of *FBN1* were located in the cbEGF region and a conserved domain across species and were not reported previously.

**Conclusions:** Our study extended and strengthened the vital role of *FBN1* LOF mutations in the pathogenesis of MFS with an autosomal dominant inheritance pattern. We confirm that genetic testing by next-generation sequencing of blood DNA can be fundamental in helping clinicians conduct mutation-based pre- and postnatal screening, genetic diagnosis and clinical management for MFS.

## INTRODUCTION

Marfan syndrome (MFS) is an inherited connective tissue disorder with autosomal dominant transmission. The clinical manifestations of MFS vary from individual to individual. More than 30 different signs and symptoms are variably associated with MFS. The most prominent of these affect the skeletal, cardiovascular and ocular systems, but all fibrous connective tissue throughout the body can be affected (*Pyeritz & McKusick, 1979*). Clinically, aortic dilatation and dissection are the most important and life-threatening manifestations of MFS (*Biggin et al., 2004*). The estimated prevalence is one in 5,000 individuals, without gender predilection (*Sponseller, Hobbs & Pyeritz, 1995*; *Von Kodolitsch & Robinson, 2007*). An epidemiological study in Taiwan revealed that the overall prevalence of MFS in Chinese population was 10.2 (95% CI [9.8–10.7]) per 100,000 individuals (*Chiu et al., 2014*).

Mutations in *FBN1* (Online Mendelian Inheritance in Man (OMIM) #134797, encoding fibrillin-1) account for 70–80% of MFS (*Stheneur et al., 2009*). In addition to *FBN1*, there are other candidate genes functionally related to MFS, such as *TGFBR1*, *TGFBR2*, *ACTA2*, *SMAD3*, *MYH11* and *MYLK*. *Habashi et al. (2006)* showed that an aortic aneurysm in a mouse model of MFS is associated with increased TGF-beta signaling and noncanonical (Smad-independent) TGF-beta signaling may be a prominent driver of aortic disease in MFS mice (*Holm et al., 2011*).

Traditionally, the discovery of pathogenic genes for MFS has depended on locus mapping using a candidate-gene strategy with family-based designs, while *FBN1* mutations have not been detected in 10% of MFS patients from clinical diagnosis, implying that either atypical mutation types or other genes may cause MFS-like disease (*Li et al., 2017*). Most cases inherit MFS from their parents in an autosomal dominant fashion (*Wieczorek et al., 1996*). MFS may also be caused by dominant negative-typemutations and haploinsufficiency (*Hilhorst-Hofstee et al., 2011*; *Judge et al., 2004*; *Judge & Dietz, 2005*). Therefore, more pathogenic genes or atypical mutations in specific populations remain to be identified. Here, we performed a family-based study using whole-exome sequencing (WES) in 19 individuals, who were derived from three Han Chinese MFS families. We identified three novel loss of function (LOF) mutations in *FBN1* likely to cause MFS in these patients. Systematical evaluations and experimental replications were conducted to validate our findings.

## MATERIALS AND METHODS

### Study subjects

A total of 19 volunteers from three Han Chinese families were recruited from the First Affiliated Hospital of Nanjing Medical University between 2012 and 2016. The mean age of onset of cases was $24.6 \pm 6.8$ years (Family 1: I-1 lost to follow-up). MFS was diagnosed through a medical record review, physical examination and family history based on Ghent nosology: (i) Ectopia lentis; (ii) Systemic score $\geq 7$; (iii) Aortic root $Z$-score $\geq 2$, when there is history of MFS in a primary relative (*Loeys et al., 2010*). The study was approved by the institutional ethical committee of Nanjing Medical University and complied with the principles of the declaration of Helsinki. Informed consent was obtained from all subjects.

## Whole-exome sequencing

Genomic DNA was isolated from peripheral blood using the QIAamp™ DNA and Blood Mini kit (Qiagen™, Munich, Germany) according to the protocol. Total DNA concentration and quantity were assessed by measuring absorbance at 260 nm with NanoDrop 2000c Spectrophotometer (Thermo Scientific™, Waltham, MA, USA). WES library construction and sequencing were performed as below: 300 ng genomic DNA was fragmented in a Covaris® M220 Focused-ultrasonicator™ to 100–500 bp fragments followed by end repair, adding A-tailing, adaptor ligation, and 11 PCR cycles according to the manufacturer's protocols. After hybridization, exome enrichment was conducted with the Agilent XT SureSelect Human All Exon v5 Kit, which targets ∼50 Mb of the human exonic regions. Five DNA libraries were multiplexed on every lane and 101 base paired-end sequencing was performed on Illumina HiSeq1500 (Illumina, Inc., San Diego, CA, USA).

## Quality control, mapping and variant calling

Raw sequencing reads were filtered to trim adapters and low quality reads using Trimmomatic-0.3.2 under PE module (ILLUMINACLIP: adapter. fa: 2:30:10; LEADING: 3; TRAILING: 3; SLIDINGWINDOW: 4:15; MINLEN: 20). All the qualified reads were processed with an in-house bioinformatics pipeline, which followed the best practice steps suggested by Genome Analysis Toolkit (GATK v3.5) (*DePristo et al., 2011*). Briefly, we first aligned the clean sequence reads to the human reference genome (UCSC Genome Browser hg19) using Burrows–Wheeler Aligner (BWA-MEM v0.7.12 with default parameters) (*Li & Durbin, 2010*). PCR duplicates were removed by Picard v1.141. After initial quality control, all eligible sequences were determined for regional realignment and base quality recalibration with GATK v3.5. Then variations including single nucleotide variants, insertions and deletions were called using GATK v3.5 HaplotypeCaller module.

## Gene classification and functional annotation

We used Annovar (2016Feb01) (*Wang, Li & Hakonarson, 2010*) for functional annotation with OMIM, the Exome Aggregation Consortium (ExAC) Browser, MutationTaster2 and The Combined Annotation Dependent Depletion (CADD). Based on OMIM and MFS-related literature reported previously, genes were classified into three categories according to American College of Medical Genetics (ACMG) standards and guidelines (*Richards et al., 2015*): Category I: eight MFS-causing genes reported directly; Category II: 125 MFS-related genes from GeneCards; Category III: Unknown genes not reported previously (Table S1).

## Manual review and replication using Sanger sequencing

All remaining mutations were manually inspected using the Integrated Genome Viewer (IGV 2.3.80) (*Thorvaldsdottir, Robinson & Mesirov, 2013*) before Sanger sequencing. PCR primers were designed for the target regions and were used to amplify these regions by PCR for Sanger sequencing. Primers are listed in the Table S2. Mutations were validated according to the resulting data screened through Chromas 2.4.1 and Dnaman 6.0.

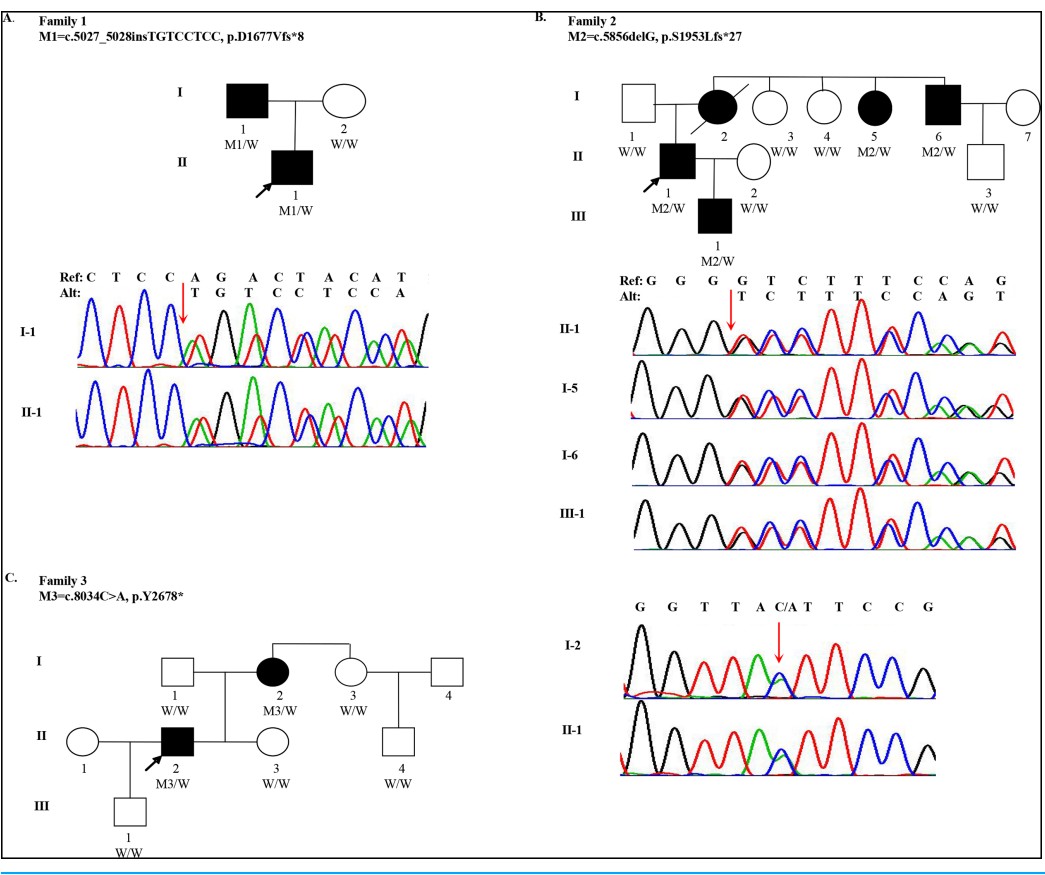

**Figure 1 Pedigree and mutations in *FBN1* for Marfan syndrome patients.** (A) A *FBN1* insertion mutation (M1) was identified in two subjects with MFS (I-1 and II-1) from Family 1; (B) A *FBN1* deletion mutation (M2) was identified in four subjects with MFS (I-5, I-6, II-1 and III-1) from Family 2; (C) A *FBN1* nonsense mutation (M3) was identified in two subjects with MFS (I-2 and II-2) from Family 3. Three individuals in the pedigrees were not sequenced including Family 2: I-2, I-7 and Family 3: I-4. W indicates wildtype allele. Circles represent female participants and squares male participants. Black symbols indicate patients with Marfan syndrome. A slash through the symbol indicates that the family member is deceased. Arrows indicate the proband.  

## RESULTS

One 18-year-old male (the proband, II-1, age of onset was 10) of Han Chinese ancestry from Family 1 was diagnosed with MFS. He presented with acute, anterior chest pain, on admission. His father (I-1) also had MFS (Fig. 1A). Computed Tomography (CT) revealed that the proband had an aortic aneurysm with dissection (type III), ascending aortic root dilatation with the diameter of 4.9 cm. Family 2 is a large three-generation family with five family members affected (I-2, I-5, I-6, II-1 and III-1) (Fig. 1B). The proband (II-1, age of onset was 30), a 31-year-old male, presented with elongated digits but no pectus excavatum. CT showed an aortic aneurysm with dissection (type I). His mother (I-2), one aunt (I-5) and one uncle (I-6) all had MFS with aortic aneurysms. His mother underwent surgery for aortic dissection in 1993 and died in 2015. Family 3 spanned three generations with two family members affected (I-2 and II-2). The proband (II-2, age of onset was 27) was a 28-year-old male with a history of MFS.

**Table 1 Clinical symptoms of all 19 members in three Marfan families.**

| Family ID | Member ID | Age of onset | Age | Wrist and thumb sign | Pectus carinatum deformity (pectus excavatum or chest asymmetry) | Hindfoot deformity (plain pes planus) | Dural ectasia | Protrusio acetabuli | Pneumo-thorax | Reduced upper segment/lower segment ratio and increased arm/height and no severe scoliosis | Scoliosis or thoracolumbar kyphosis | Reduced elbow extension | Facial features | Skin striae (stretch marks) | Myopia > 3 diopters | Mitral valve prolapse | Systemic score | Aortic root Z-score | Case |
|---|---|---|---|---|---|---|---|---|---|---|---|---|---|---|---|---|---|---|---|
| Score | | | | 3 | 2(1) | 2(1) | 2 | 2 | 2 | 1 | 1 | 1 | 1 | 1 | 1 | 1 | | | |
| F1 | I-1 | 25 | 46 | √ | √ | × | × | × | √ | × | × | × | × | × | × | √ | 8 | ND | 1 |
| F1 | I-2 | | 43 | × | × | × | × | × | × | × | × | × | × | × | × | × | 0 | ND | 0 |
| F1 | II-1 | 10 | 21 | √ | √ | × | × | × | √ | × | × | × | × | × | × | √ | 8 | ≥2 | 1 |
| F2 | I-1 | | 66 | × | × | × | × | × | × | × | × | × | × | × | × | × | 0 | ND | 0 |
| F2 | I-3 | | 60 | × | × | × | × | × | × | × | × | × | × | × | × | × | 0 | ND | 0 |
| F2 | I-4 | | 63 | × | × | × | × | × | × | × | × | × | × | × | × | × | 0 | ND | 0 |
| F2 | I-5 | 30 | 65 | × | × | √ | × | × | × | √ | × | √ | √ | × | × | √ | 6 | ND | 1 |
| F2 | I-6 | 20 | 67 | × | × | √ | × | × | × | √ | × | √ | √ | × | × | √ | 6 | ND | 1 |
| F2 | II-1 | 30 | 36 | × | × | √ | × | × | × | √ | × | √ | √ | × | × | √ | 6 | ≥2 | 1 |
| F2 | II-2 | | 30 | × | × | × | × | × | × | × | × | × | × | × | × | × | 0 | ND | 0 |
| F2 | II-3 | | 50 | × | × | × | × | × | × | × | × | × | × | × | × | × | 0 | ND | 0 |
| F2 | III-1* | 2 | 5 | × | √ | × | × | × | × | × | × | × | × | × | × | × | 1 | ND | 0* |
| F3 | I-1 | | 52 | × | × | × | × | × | × | × | × | × | × | × | × | × | 0 | ND | 0 |
| F3 | I-2 | 30 | 49 | × | √ | √ | × | × | × | × | × | × | × | × | × | √ | 5 | ND | 1 |
| F3 | I-3 | | 46 | √ | √ | × | × | × | × | × | × | × | × | × | × | × | 4 | <2 | 0 |
| F3 | II-2 | 27 | 29 | × | √ | √ | × | × | × | × | × | × | × | × | × | √ | 5 | ≥2 | 1 |
| F3 | II-3 | | 28 | × | × | × | × | × | × | × | × | × | × | × | × | × | 0 | ND | 0 |
| F3 | II-4 | | 22 | × | × | × | × | × | × | × | × | × | × | × | × | × | 0 | ND | 0 |
| F3 | III-1 | | 6 | × | × | × | × | × | × | × | × | × | × | × | × | × | 0 | ND | 0 |

**Notes:**

Facial features (3/5) = 1 (dolichocephaly, enophthalmos, downslanting palpebral fissures, malar hypoplasia, retrognathia).

ND, not detected; NA, not available.

\* Suspected case.

**Table 2** *FBN1* variants identified for affected individuals in three Marfan families.

| Family ID | F1 | F2 | F3 |
|---|---|---|---|
| Chr. | chr15 | chr15 | chr15 |
| Position | 48,756,133 | 48,737,634 | 48,707,750 |
| Ref allele | – | G | C |
| Alt allele | TGTCCTCC | – | A |
| Gene | *FBN1* | *FBN1* | *FBN1* |
| Mutation type | insertion | deletion | nonsense |
| Exon | 41/66 | 48/66 | 64/66 |
| Codon change | c.5027_5028insTGTCCTCC | c.5856delG | c.8034C > A |
| Amino acid change | p.D1677Vfs*8 | p.S1953Lfs*27 | p.Y2678* |
| Affected individuals | I-1/II-1 | I-5/I-6/II-1/III-1 | I-2/II-2 |
| CADD raw score | 9.18 | 7 | 16.63 |
| PHRED scaled score[†] | 35 | 33 | 56 |

**Note:**
[†] PHRED-like scaled *C*-scores = $-10^*\log\_10$ (rank/total), the recommended deleterious threshold was >15 for scaled *C*-scores.

He had thoracic surgery for pectus excavatum at 2 years old. Although he had no clinical symptoms in the cardiovascular system, a CT scan showed ascending aortic dilatation, aortic regurgitation and mitral regurgitation. His mother (I-2) was also confirmed to have MFS (Fig. 1C). Although case I-3 presented elongation of fingers and mild pectus excavatum, there were not sufficient clinical features to perform the diagnosis (systemic score = 4 and aortic root *Z*-score < 2) (Table 1).

Quality summaries from sequencing of the 19 samples are summarized in Table S3. Each sample had an average of 69.46M raw reads, and over 99.60% of them were successfully mapped to the reference genome. The average of median insert size was 201 bp and percent of duplicate reads ranged from 1.44% to 8.11%. Totally 237,252 variants were kept for following evaluation. To identify qualified pathogenic mutations, stringent criteria according to ACMG guidelines were performed (Fig. S1). First, we filtered these variants under the following criteria: (i) untranslated region, synonymous, intronic variants (except variants considered to be splicing variants and located at exon-intron junctions ranging from −5 to +5); (ii) variants with minor allele frequency ≥ 1% based on 1,000 Genomes (1KG) and ExAC databases; (iii) variants present in our in-house whole genome sequencing database of 100 non-Marfan controls. Then, we classified these rare genetic variants into three categories: (a) MFS-causing genes; (b) MFS-related genes; (c) Unknown genes. Then we assessed whether these variants were loss-of-function (nonsense, frameshift and essential splice-site variants). Three inheritance patterns were evaluated for the qualified mutations including autosomal dominant, autosomal recessive and compound heterozygotes. Finally, we manually reviewed and selected variants in Category I and II genes for validation. After replication by Sanger sequencing, three LOF mutations in *FBN1* were identified in three families, respectively. For Family 1, the insertion (c.5027_5028insTGTCCTCC) was detected in both I-1 and II-1 individuals, which resulted in a frameshift (p.D1677Vfs*8). For Family 2, a heterozygous deletion

(c.5856delG) in exon 48 (NM_000138) was found in four patients (I-5, I-6, II-1 and III-1), also resulting in a frameshift (p.S1953Lfs*27). I-2 and II-2 individuals from family 3 were heterozygous for the nonsense variant (c.8034C>A), which gained an immediate stop codon (p.Y2678*). All mutations (c.5027_5028insTGTCCTCC, c.5856delG and c.8034C>A) identified in three families were predicted to be disease-causing using MutationTaster2 and CADD. In addition, all mutation sites were located in a highly conserved amino acid region (Calcium-binding epidermal growth factor (EGF) domain) across different species (Fig. S2). A summary of these mutations is presented in Table 2. All healthy family members and 100 other non-MFS controls did not carry these *FBN1* variants. It is worth noting that these mutations in *FBN1* have not been reported previously, but the variant (c.5857dupT), near c.5856delG, was recorded by the Human Gene Mutation Database (HGMD).

## DISCUSSION

Marfan syndrome is a systemic disorder of connective tissue with a high degree of clinical variability that involves skeletal, ocular and cardiovascular systems (*Dietz, 1993*). In our study, massively parallel sequencing was performed to identify genetic abnormalities in three MFS families, showing three rare functional variations in *FBN1*.

Fibrillins are the major components of microfibrils in the extracellular matrix of elastic and non-elastic tissues. Fibrillin-1 serves as a structural component of calcium-binding microfibrils and is encoded by *FBN1* gene. *FBN1* is mapped to chromosome 15q21.1 and encodes a 2,871 amino acid protein. More than 1,800 different mutations have been identified in *FBN1*, most of which are associated with MFS, as seen in the Universal Mutation Database (UMD)-*FBN1* mutations database (*Collod-Beroud et al., 1997*; *Collod et al., 1996*) using a generic software called UMD (*Collod-Beroud et al., 2003*). Similar to this curated database, approximately 1,700 variants in *FBN1* are associated with MFS according to the professional version of the HGMD, and these accounts for more than 90% of MFS cases. In our study, all affected individuals also carried *FBN1* LOF mutations (c.5027_5028insTGTCCTCC, c.5856delG and c.8034C>A).

The mutated mRNAs in our study, although introducing premature termination codons (PTCs), could in theory be stable. However, cellular recognition and degradation of mRNA that contains PTC via nonsense-mediated mRNA decay (NMD) is a process whereby potentially harmful effects of truncated proteins may be limited (*Culbertson, 1999*; *Frischmeyer & Dietz, 1999*). Accordingly to one study, in the majority of *FBN1* PTC MFS cases, synthesis of normal-sized fibrillin protein was ~50% of control levels, but much less matrix deposition occurred (*Schrijver et al., 2002*). They concluded that most PTC mutations have a major impact on the pathogenesis of type 1 fibrillinopathies but that it is not always through NMD. In our study, we found that two PTC mutations (p.S1953Lfs*27 and p.Y2678*) were quite near the mutations (p.Q1955X and p.I2681X) reported by *Schrijver et al. (2002)*. In contrast, the relative amount of PTC-containing *FBN1* transcripts in blood was found to be significantly higher than in affected fibroblasts with experimental inhibition of nonsense-mediated decay, while in fibroblasts without

NMD inhibition, no mutant alleles could be detected at all (*Magyar et al., 2009*), implying that tissue-specific degradation of transcripts also plays an important role in MFS pathogenesis.

Along these lines, *Faivre et al. (2007)* found that patients with an *FBN1* PTC had a more severe skeletal and skin phenotype than did patients with an in-frame mutation. Mutations in exons 24–32 were also associated with a more severe and complete phenotype. In our study, the LOF mutations were located in exon 41, 48 and 64, and patients did not have a complete MFS phenotype including ectopia lentis and skin striae. This expression of the MFS phenotype may also depend on the different ethnicity of our patient cohort from the above-cited studies.

In Family 2, a suspected case of a 4-year-old boy (patient III-1) who had longer finger and anterior chest deformity (pectus excavatum), was slightly taller than his peers. It was difficult to make a clinical diagnosis owing to his age and uncertain status according to the clinical criteria, but our WES-based screening helped ease his diagnosis by excluding the *FBN1* mutation found in other affected members of his family. In Family 3, we detected that clinically unaffected subject I-3 had a slight anterior chest deformity (pectus excavatum). Although she had this MFS-related symptom, she was clinically considered as a healthy individual, which was borne out by the result that she had no *FBN1* or other pathogenic mutations.

All family members above are followed up regularly to confirm their diagnoses. The identification of a causative gene variant by WES in those with an uncertain phenotype or complex subjects is of inestimable value for screening, clinical diagnosis and, ultimately, directing personalized patient care with development of specific small-molecule therapies.

## CONCLUSIONS

In conclusion, our results may help further elucidate the genetic pathology of MFS, and these mutations could be included among probably pathogenic markers for pre- and postnatal screening and genetic diagnosis for MFS.

## WEB RESOURCES

Qiagen™, https://www.qiagen.com
Agilent™, https://www.agilent.com
Trimmomatic-0.3.2, http://www.usadellab.org/cms/index.php?page=trimmomatic
Genome Analysis Toolkit (GATK v3.5), https://software.broadinstitute.org/gatk
Burrows–Wheeler Aligner, BWA v0.7.12, http://bio-bwa.sourceforge.net
Picard v1.141, http://picard.sourceforge.net
Annovar (2016Feb01), http://annovar.openbioinformatics.org
Mendelian Inheritance in Man (OMIM), http://www.omim.org
Exome Aggregation Consortium (ExAC) Browser, http://exac.broadinstitute.org
MutationTaster2, http://www.mutationtaster.org/
The Combined Annotation Dependent Depletion (CADD), http://cadd.gs.washington.edu
GeneCards, https://www.genecards.org/Search/Keyword?queryString=marfan%20syndrom

Integrated Genome Viewer (IGV 2.3.80), http://software.broadinstitute.org/software/igv/
Human Gene Mutation Database (HGMD), http://www.hgmd.cf.ac.uk
Universal Mutation Database (UMD), http://www.umd.be

### Funding

This work was supported by the Natural Science Foundation of Jiangsu Province (BK20151590), the Priority Academic Program Development of Jiangsu Higher Education Institutions (Public Health and Preventive Medicine) and Top-notch Academic Programs Project of Jiangsu Higher Education Institutions (PPZY2015A067). The funders had no role in study design, data collection and analysis, decision to publish, or preparation of the manuscript.

### Grant Disclosures

The following grant information was disclosed by the authors:
Natural Science Foundation of Jiangsu Province: BK20151590.
Priority Academic Program Development of Jiangsu Higher Education Institutions (Public Health and Preventive Medicine).
Top-notch Academic Programs Project of Jiangsu Higher Education Institutions: PPZY2015A067.

### Competing Interests

The authors declare that they have no competing interests.

### Author Contributions

- Zhening Pu analyzed the data, prepared figures and/or tables, authored or reviewed drafts of the paper, approved the final draft.
- Haoliang Sun analyzed the data, contributed reagents/materials/analysis tools.
- Junjie Du analyzed the data, contributed reagents/materials/analysis tools.
- Yue Cheng performed the experiments.
- Keshuai He performed the experiments.
- Buqing Ni performed the experiments.
- Weidong Gu performed the experiments.
- Juncheng Dai conceived and designed the experiments, authored or reviewed drafts of the paper, approved the final draft.
- Yongfeng Shao conceived and designed the experiments, authored or reviewed drafts of the paper, approved the final draft.

### Human Ethics

The following information was supplied relating to ethical approvals (i.e., approving body and any reference numbers):

The study was approved by the institutional ethical committee of Nanjing Medical University and complied with the principles of the declaration of Helsinki.

## Data Availability

The raw data are provided in the Tables, Figures and Supplemental Files.

## Supplemental Information

Supplemental information for this article can be found online at http://dx.doi.org/10.7717/peerj.5927#supplemental-information.

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
