# Peer review of "Family-based whole-exome sequencing identifies novel loss-of-function mutations of FBN1 for Marfan syndrome"

_PeerJ, doi:10.7717/peerj.5927_

## Round 0.1 · original submission · Major Revisions

· Academic Editor

Major Revisions

We have now received feedback from three expert reviewers who have made a number of suggestions to improve this work. As PeerJ selects articles based only on a determination of scientific and methodological soundness, not on subjective determinations of 'impact,' 'novelty' or 'interest', we would nonetheless encourage you to take each of the reviewers' suggestions and comments into account before re-submitting your work. Please address them each point by point in your letter of response.

One of the important rectifications to include in your revised submission concerns a table of the clinical features of the patients and a correspondence between these detailed phenotypes and their genotypes, including apparently non-pathogenic variants discovered in the genes assessed. Another commonly suggested revision concerns the current absence of discussion of NMD mechanisms.

We concur with the assessment that the data you report could be valuable to the scientific community and look forward to receiving a carefully revised manuscript that would communicate their interest.

·

Basic reporting

No comment

Experimental design

No comment

Validity of the findings

No comment

Additional comments

In this manuscript, Pu et al. describe 3 FBN1 gene mutations (c.5027_5028insTGTCCTCC, c.5856delG and c.8034C>A) identified by next generation sequencing in 3 Han Chinese families. These 3 mutations result in premature STOP codons. These mutations are unreported in large public databases gathering variants identified by exome and genome sequencing as gnomAD. These mutations have not been previously reported in Chinese patients (more than 150 reported to date) nor in the FBN1 locus specific databases UMD-FBN1. The manuscript is clear and concise, and results are presented in a manner that is logical.

I do regret, however, the absence of clear clinical description:
- line 126: “…diagnosed with MFS…”
- line 127: “His father (I-1) also had MFS…”
- line 132-133: ”…all had MFS with…”.
I suggest adding a table with a complete and clear description of all symptoms (skeletal, ocular and cardiovascular systems as well as other associated systems) for each of the 23 members of these families.

The second important point is in the way the authors discuss the pathogenicity of these mutations (lines 192 to 205). Even though these affirmations are right, the major point for each of these 3 mutations is the creation of premature STOP codons. The question is then about the activation of nonsense mediated decay (mRNA degradation and resultant haploinsufficiency) or the production of truncated proteins.
Are the mutated mRNAs stable?
Are the mutant proteins detected?

[For more information please refer to:
- Schrijver I, et al. Premature termination mutations in FBN1: distinct effects on differential allelic expression and on protein and clinical phenotypes. Am J Hum Genet. 2002;71:223–237. (PMID: 12068374).
- Tjeldhorn L, Amundsen SS, Barøy T, Rand-Hendriksen S, Geiran O, Frengen E, Paus B.
Qualitative and quantitative analysis of FBN1 mRNA from 16 patients with Marfan Syndrome. BMC Med Genet. 2015 18;16:113. PMID: 26684006
- Faivre L, et al. Effect of mutation type and location on clinical outcome in 1,013 probands with Marfan syndrome or related phenotypes and FBN1 mutations: an international study. Am J Hum Genet. 2007;81:454–466. (PMID: 17701892).]

Publication of such data is important as they often lie within clinical laboratories and are not shared with the medical and research communities. The clinical spectrum associated with FBN1 gene mutations is very large and complex. A lot of data has then to be colligated to identify genotype-phenotype and gene-function correlations.

Reviewer 2 ·

Basic reporting

This study is pretty straight forward and authors followed a standard procedure to identify mutations in FBN1 lead to MFS using whole exome sequencing.

Experimental design

The methods described provide enough detail information for replication. The question of this study not clear. It seems that authors don’t familiar with the rule of mammalian nonsense mediated mRNA decay due to frameshift or stop-gain variants. This study doesn't provide sufficient novelty to fill knowledge gap.

Validity of the findings

The impact and novelty are low.

Additional comments

In this manuscript, authors performed whole-exome sequencing on patients with Marfan syndrome (MFS) from 19 families and identified novel loss-of –function FBN1 variants in three families, two frameshift and one stop-gain variants. These variants were segregated with MFS in families. Authors tried to investigate the role of novel genetic variants in determining MFS clinical phenotype. Authors have discussed the domain location and local conservation of these variants in FBN1 and compared the diversity of clinical status of patients.
Concerns:
1. All of these variants lead to haplo-insufficiency of FBN1 and individuals carried these mutations should have similar clinical features with MFS based on the clinical criteria MFS diagnosis. The pathogenic variants lead to haplo-insufficiency of FBN1 were frequently seen in patients with MFS. It seems that authors don’t familiar with the rule of mammalian nonsense mediated mRNA decay due to frameshift or stop-gain variants.
2. There is insufficient novelty or significance of this manuscript to meet publication criteria in Peer J.

Reviewer 3 ·

Basic reporting

English. Unclear/wrong sentences or words are present throughout the article
Row 42: “major features”. Why major features?
Row 51: “in the genes” , only one gene
Row 62: ”relate to the dominant negative”….
Rows 63-64: “Therefore…….MFS”. Marfan syndrome is at the moment associated
with FBN1 gene only. What do you mean with “atypical mutation”?.
Rows 104-105 “Then……module”. Unclear sentence
Rows 136-137 “Although…..symptoms” What the Authors mean by “obvious
symptoms”?
Rows 139-140: “elongation of fingers” = arachnodactily; what the Authors mean
by “were attributed to being healthy”?
Row 146: “we removed these variants”
Rows 152-153 “MFS……….function” english and meaning are both unclear
Row 158 “a recurrent insertion”. What the Authors mean by recurrent? They said
previously that it was a “new muation”
Row 176 “several functional mutations”…..All rare mutations” Are the Authors
talking about the three FBN 1 mutations detected in the 3 families or
about other mutations?
Row 187-189. English and meaningunclear
Row 198-200 . English and meaning unclear
Row 206-207 . English and meaning unclear




Literature references.
References are sufficient. Within the manuscript Reinhardt 2015 is reported buti s omissed in the reference section.


Background.
The background is not very updated. The Author scite many genes associated with marfan syndrome but actually only one, the FBN1 gene, is associated with it.
The Authors mention a “a complex pathogenesis underlying MFS” but do not mention the up-regulation mechanism of TGF-beta signalling.
The Authors use the term “major features” that has disappeared in the latest guidelines (Loeys et al 2010), substituted by the term “features characterizing the disease”. Moreover they mention only the three clinical manifrstations by omissing the two genetic features
Finally, it is not clear why they emphasize the fact thet Marfan patients carry dominant negative or haploinsufficient mutations.

Professional article structure, figs, tables. Raw data shared.
An important table is missing: the one with all the clinical manifestations (such as myopia, mitral valve prolapse, scoliosis, ectopia lentis) regarding the 19 subjects of the three families. Moreover, the Authors should report the cardiovascular manifestations (such as hypertension and dyslipidemias) which can modulate some clinical features of the patients.
Moreover, a second Table should report all the mutations detected in the genes associated to TGFbetapathies or thoracic aortic aneurysms/dissections because we know from the literature that milt mutations in these genes may rapresent modifyer genes of Marfan syndrome. The Authors do not mention modifier genes in the background section.


Self-contained with relevant results to the hypotheses.
The Authors aim to investigate the role of novel genetic variants in determining clinical phenotype.
With this aim I expected a deep presentation of the clinical features detected in all 19 subjects, includin healthy relatives. This is the only way to perform a genotype-phenotype correlation. Therefore this reviewer thinks that important are missing. On the other side they analyzed many genes potentially modifier genes of Marfan syndrome but no data are reported. Also these data would allow improve a genotype-phenotype correlation.
Furtheremore, it is difficult to believe that the Authors found only one mutation in FBN1 gene in each patient. FBN1 not-pathogenic variants should also be reported in a third Table.

Experimental design

Study subject. The age of the 19 subjects should be reported here or in the pedigree.

Methods. Sufficient.

Validity of the findings

The Authors report 3 new mutations all causin LOF = loss of function. Since they have 3 families they shoud concentrate result and discussion on genotype-phenotype correlation by reporting al the mutations detected in the other genes and all clinical features of the 19 subjectes. Being only 3 mutations I would avoid a comparison of these mutations functions with those reported in literature.


Results.
The resultus are confusing since contain in part information regarding methods.

Discussion.
The discussion is poor and unclear/confused.
The diagnosis of Marfan syndrome is still based primarily on clinical manifestations since in part of Marfan patients we still do not detect a pathogenic mutation. Of course the age of diagnosis is around 20 years because earlier most of the times sufficient clinical features necessary to perform the diagnosis are still missing. A correct follow up requires early diagnosis

Additional comments

The Authors performed whole-exome sequencing on 19 individuals from 3 Han Chinese families affected by Marfan syndrome. Their purpose was to investigate the role of novel genetic variants in determining Marfan syndrome as reported in the ‘Background” of the summary.
The article needs deep revision. Clinical and genetic data necessary for genotype phenotype correlation are missing at present.

---

## Round 0.2 · Minor Revisions

· Academic Editor

Minor Revisions

The addition of the clinical table has helped. Please re-examine these remarks by a reviewer that appear to remain unaddressed from the previous revision: "The second important point is in the way the authors discuss the pathogenicity of these mutations (lines 192 to 205 [first submission line numbers]). Even though these affirmations are right, the major point for each of these 3 mutations is the creation of premature STOP codons. The question is then about the activation of nonsense mediated decay (mRNA degradation and resultant haploinsufficiency) or the production of truncated proteins. Are the mutated mRNAs stable? Are the mutant proteins detected?"

These remarks are echoed in the re-review that was returned by one of the reviewers. These need discussion and possibly some minimal experimentation to address.

The English language is sufficient to communicate with non-English speakers but is still not correct. Please have your manuscript revised by someone whose mother tongue is English, as there remain a number of grammatical mistakes.

Reviewer 2 ·

Basic reporting

The article written in English is clear.
The sufficient introduction and background are provided.
The structure of figures and tables are fine.

Experimental design

The experimental design is fine. It is a standard exome data analysis approach.

Validity of the findings

Impact and novelty are low.
The conclusions are appropriately stated,

Additional comments

In this manuscript, authors performed whole-exome sequencing on patients from 3 families with Marfan syndrome (MFS) and identified novel loss-of –function FBN1 variants, two frameshift and one stop-gain variants. These variants lead to premature termination of FBN1 and therefore, they are pathogenic variants for MFS. Authors have addressed some questions that brought up by reviewers. However, there are still some concerns about this manuscript.

Concerns:
1. This manuscript lacks sufficient novelty or significance information to readers. In the abstract, authors purposed to investigate the role of novel genetic variants in determining MFS clinical phenotype. However, a lot of FBN1 pathogenic variants that lead to haplo-insufficiency of FBN1 have been reported and the patients’ clinical phenotypes have been well studied. It may be interested to see whether the clinical phenotypes are difference between Chinese patients with FBN1 PTC mutations and that of patients of European American descent.
In addition, in the discussion, authors propose to conduct protein truncation test to confirm the activation of nonsense mediated decay or the production of truncated proteins. However, such studies cannot generate any novel information for FBN1 PTC mutations because they have been well studied.
2. Since all of the mutations identified in this study lead to premature termination of FBN1, the discussion of local conservation of the amino acids mutated is not necessary and the figure 2 can be removed.
3. The FBN1 variants identified in these families are definitely the causative variants for MFS in these families. There are no evidence that mutations in FBN3 lead to MFS (JACC 72(6):605). Therefore, it is not necessary to describe the FBN3 variants.

---

## Round 0.3 · Minor Revisions

· Academic Editor

Minor Revisions

I found it easier to make the corrections directly on the manuscript. Please find it attached. Also, please remove the reference to the textbook as superfluous. If you had needed it, it is always preferable to use the primary literature. You will still need to copy-edit later on. Otherwise, I do find the manuscript acceptable for publication if you make these corrections (or convincingly justify why you choose not to). Thank you for submitting your paper to PeerJ and for your contributions to the study of MFS around the world.

---

## Round 0.4 · accepted · Accept

· Academic Editor

Accept

I see that my suggestions for the previous revision have been incorporated and am glad to accept this version.

#